# Insulin Can Delay Neutrophil Extracellular Trap Formation In Vitro—Implication for Diabetic Wound Care?

**DOI:** 10.3390/biology12081082

**Published:** 2023-08-03

**Authors:** Caren Linnemann, Filiz Şahin, Ningna Li, Stefan Pscherer, Friedrich Götz, Tina Histing, Andreas K. Nussler, Sabrina Ehnert

**Affiliations:** 1Siegfried Weller Institute for Trauma Research, BG Unfallklinik Tübingen, Eberhard Karls Universität Tuebingen, 72076 Tuebingen, Germany; caren.linnemann@med.uni-tuebingen.de (C.L.); andreas.nuessler@gmail.com (A.K.N.); 2Microbial Genetics, Interfaculty Institute of Microbiology and Infection Medicine, Eberhard Karls Universität Tuebingen, 72076 Tuebingen, Germany; 3Department of Internal Medicine III, Sophien- and Hufeland-Hospital, 99425 Weimar, Germany

**Keywords:** insulin, neutrophil extracellular traps, wound healing, diabetes

## Abstract

**Simple Summary:**

Diabetes is a globally developing disease. Diabetic patients suffer from several side diseases, including impaired healing abilities. In general, every injury is characterized by an immediate immune response at the wound site. Neutrophils are the first immune cells arriving at the fracture site, to defend the body against pathogens. However, neutrophils may also induce tissue damage. By releasing their DNA together with proteins, so-called neutrophil extracellular traps (NETs), they can induce an overshooting inflammation and directly harm tissue, especially during wound healing. Some anti-diabetic drugs were already found to influence this process but for one of the major treatments of diabetes, insulin, nothing is known so far. We found that insulin can modulate the process of NET formation and instead induce the anti-bacterial capacity of neutrophils. Thus, insulin may be a tool to improve diabetic wound healing by regulation of NET formation.

**Abstract:**

Diabetes is a worldwide evolving disease with many associated complications, one of which is delayed or impaired wound healing. Appropriate wound healing strongly relies on the inflammatory reaction directly after injury, which is often altered in diabetic wound healing. After an injury, neutrophils are the first cells to enter the wound site. They have a special defense mechanism, neutrophil extracellular traps (NETs), consisting of released DNA coated with antimicrobial proteins and histones. Despite being a powerful weapon against pathogens, NETs were shown to contribute to impaired wound healing in diabetic mice and are associated with amputations in diabetic foot ulcer patients. The anti-diabetic drugs metformin and liraglutide have already been shown to regulate NET formation. In this study, the effect of insulin was investigated. NET formation after stimulation with PMA (phorbol myristate acetate), LPS (lipopolysaccharide), or calcium ionophore (CI) in the presence/absence of insulin was analyzed. Insulin led to a robust delay of LPS- and PMA-induced NET formation but had no effect on CI-induced NET formation. Mechanistically, insulin induced reactive oxygen species, phosphorylated p38, and ERK, but reduced citrullination of histone H3. Instead, bacterial killing was induced. Insulin might therefore be a new tool for the regulation of NET formation during diabetic wound healing, either in a systemic or topical application.

## 1. Introduction

Diabetes mellitus is a disease with a rapidly increasing prevalence and incidence worldwide [1]. More than 400 million people are expected to be diagnosed with diabetes in 2030, with the most rapidly increasing numbers in low- and middle-income countries [1]. Type 2 diabetes mellitus (T2DM) is characterized by elevated blood glucose levels and insulin resistance (accompanied by increased insulin secretion at the onset of the disease) and has severe effects on the entire body. In addition to nephro-, neuro-, and angiopathies, healing processes are generally impaired in T2DM [2]. Delayed wound healing, chronic wounds, and subsequent amputations are major clinical problems associated with high medical costs and great suffering for patients [3]. Diabetes still is the major cause of non-traumatic amputations worldwide [4]. The wound healing itself is a highly complex and timely orchestrated process that frequently fails. Several factors are thought to play a role, including angiopathy, neuropathy, tissue remodeling, or an altered immune response [2].

The latter is mainly important directly after the injury, when inflammatory cells invade the wound area. Prolonged or dysregulated inflammation is associated with a poor healing outcome [5]. Neutrophils, which are the first immune cells arriving at the wound site [6], are usually responsible for eliminating pathogens in the wound. Moreover, they are responsible for attracting additional immune cells and other cell types, such as, e.g., fibroblasts, to the wound site [7]. Recently, they were found to release their DNA as a fast and unspecific defense mechanism against pathogens. In combination with histones and anti-microbial proteins, the released DNA forms large extracellular structures, the so-called neutrophil extracellular traps (NETs). Although they are an effective weapon against pathogens, NETs have also been found to be involved in delayed wound healing (reviewed in [8]), particularly in diabetic wounds [9,10]. Increased levels of citrullinated histone H3, a marker for NET release, were associated with amputations in diabetic foot patients [10]. Consequently, inhibition of NET formation improved wound healing in diabetic mice [9]. 

Diabetic patients are not only affected by the disease itself but also by the prescribed medication, which may also affect the wound-healing process. One of the first choices in the treatment of type 2 diabetes is metformin [11]. It efficiently reduces hyperglycemia and was shown to have a positive effect on wound healing in non-diabetic mice through an AMPK (AMP-activated protein kinase)-dependent mechanism [12]. In diabetic mice, metformin also improved wound angiogenesis and reversed endothelial cell dysfunction [13]. In addition to its effects on wound healing in mice, metformin has been shown to reduce NET formation in T2DM patients [14,15] and in neutrophils in vitro [14,16]. A relatively new generation of anti-diabetic drugs are the incretin mimetics such as GLP-1 (glucagon-like peptide 1) analogs. The GLP-1 analog liraglutide has been shown to inhibit NET formation in a mouse cancer model [17], indicating that not only metformin could regulate NET formation. 

When blood glucose is difficult to control, insulin therapy is recommended early in T2DM treatment [18]. Insulin not only has blood sugar-regulating abilities but can also act as a growth factor or as an immunoregulator [19], and has been shown to downregulate myeloperoxidase (MPO) activity in mice [20]. This particular effect may account for the acceleration of wound closure observed in diabetic mouse models [21] and small clinical trials [22,23]. However, to our knowledge, the direct effect of insulin on NET formation has not yet been investigated. Thus, in this study, the effect of insulin on NET formation in response to various NET stimuli was investigated. 

## 2. Materials and Methods

### 2.1. Human Material

Venous blood was collected from healthy volunteers after informed consent according to the Declaration of Helsinki. The study was approved by the Ethics Committee of the University of Tuebingen (Ethical vote: 666/2018B02). 

### 2.2. Neutrophil Isolation

Neutrophils were isolated from venous blood as previously described [24]. Briefly, equal amounts of fresh blood and lympholyte poly isolation medium (Cedarlane, Burlington, ON, Canada) were layered. After centrifugation at 500× *g* for 35 min without break, the polymorphonuclear cell (PMN) layer was collected and washed twice with PBS. The plasma layer was saved and kept on ice until use. Neutrophils were resuspended in plain RPMI medium (Sigma-Aldrich, St. Louis, MO, USA) and counted by the trypan blue exclusion method without counting residual erythrocytes. All experiments were performed with the addition of 2% autologous plasma, only with lipopolysaccharide (LPS) stimulation, no plasma was added (LPS only induced NET release without plasma, see Appendix A). 

### 2.3. Diabetic Conditions

The standard concentration used for stimulation with insulin was 160 IU/L insulin ([Ins], Actrapid, Novo Nordisk, Copenhagen, Denmark) [25]). High glucose (HG) conditions were simulated with 25 mM glucose (Sigma-Aldrich). A combination of glucose and insulin is indicated as HG Ins. HG +Ins was added 1 h before the actual stimulation to pre-stimulate the cells with diabetic conditions, Ins alone was directly added with the stimulants (phorbol myristate acetate [PMA], LPS). Figure 1 gives an overview of the experimental set-up. 

### 2.4. Induction of NETs

#### 2.4.1. Chemicals and Peptides for Stimulation

PMA (100 nM), CI (Calcium ionophore A23187, 4 µM), and LPS (from *Escherichia coli* O111:B4, 25 µg/mL) were used as stimuli.

#### 2.4.2. Bacteria for Stimulation

For the stimulation of neutrophils for NET release in the presence of bacteria, the following four bacterial strains that are frequently found in infections were used [26,27,28]: *Staphylococcus aureus* (*S. aureus*, *SA*), *Staphylococcus epidermis* (*S. epidermis*, *SE*), *Pseudomonas aeruginosa* (*P. aeruginosa*, *PA*), *Enterococcus faecalis* (*E. faecalis*, *EF*).

*S. aureus* USA300 JE2, *S. epidermidis* O47, and *E. faecalis* DSM20478 were grown in trypticase soy broth (TSB) medium at 37 °C, while *P. aeruginosa* PAO1 were grown in lysogeny broth (LB) medium. The overnight cultures were diluted to an optical density (OD_578_) of 0.1, and incubated at 37 °C with shaking at 120 rpm for 2 h. The bacteria were then centrifuged at 4700× *g* for 10 min, and washed with PBS. Bacterial dosage (MOI, multiplicity of infection) was defined by the colony-forming units (CFUs) per OD_578_. For *S. aureus* JE2, *S. epidermidis*, and *E. faecalis*, an OD_578_ of 0.3 equals 1 × 10^8^ CFU/mL, however, for *P. aeruginosa*, an OD_578_ of 0.1 equals 1 × 10^8^ CFU/mL.

Neutrophils were co-incubated with bacteria (MOI 50) with or without addition of insulin (160 IU/L). NET formation was analyzed by Sytox Green Assay for 5 h. Afterward, the culture supernatants were collected and plated on TSR agar plates in a serial dilution (10^−3^ to 10^−5^). After overnight incubation at 37 °C, CFUs were counted to determine bacterial survival after co-incubation with neutrophils.

### 2.5. Analysis of NET Formation

The formation of NETs was analyzed by two different methods. For initial screening, the Sytox Green assay was used, with further confirmation made by immunofluorescence staining.

#### 2.5.1. Sytox Green Assay

Neutrophils were diluted to 2 × 10^5^ cells/mL. 1 µM Sytox Green (Thermo Fisher, Waltham, MA, USA) and stimulants were added directly and green fluorescence was measured every 30 min with the ClarioStar plate reader (BMG Labtech, Ortenberg, Germany) at 37 °C, 5% CO_2_. Neutrophils lysed with 1% Triton-X-100 were defined as 100% DNA release and fluorescence values normalized to that value. As readout parameters, the area under the curve was calculated to determine the total DNA release over time and the EC_50_ values were used as half-maximal stimulation time to analyze the dynamics of DNA release [24]. 

#### 2.5.2. Immunofluorescence

The analysis of NET formation by immunofluorescence was performed as previously described [24]. Cells were diluted to 3 × 10^5^ cells/mL and seeded in self-made poly-L-lysine coated chamber slides. Stimulants were added directly, and the cells were incubated at 37 °C and 5% CO_2_. The stimulation times were taken from the results of the Sytox Green assay. Cells were fixed with 4% formaldehyde and permeabilized with 0.5% Triton-X-100. After blocking with 5% bovine serum albumin (BSA) in PBS, cells were incubated overnight with myeloperoxidase antibody (1:200 in PBS, sc-52707, Santa Cruz Biotechnology, Heidelberg, Germany). After washing with PBS, the staining was continued with Alexa Fluor-488 coupled secondary antibody (1:1000 in PBS, #A10667, Invitrogen, Carlsbad, CA, USA) and Hoechst 33342 (2 µg/mL) for 2 h. The chambers were removed, and the slides were mounted (Fluoromount G mounting medium, Thermo Fisher) and covered with a cover glass. Microscopy was performed with an EVOS Fl fluorescence microscope (Thermo Fisher). From each well, at least five images were taken and analyzed for NET formation with ImageJ (Version 1.53) as described [24]. NET formation is given as the percentage of NETosed cells. 

### 2.6. Reactive Oxygen Species

Neutrophils were diluted to 1 × 10^6^ cells/mL and subsequently stained with the reactive-oxygen-species-sensitive dyes for 20 min at 37 °C, 5% CO_2_. After washing once with PBS, stimulants were added, and fluorescence was directly measured in an OmegaPlate reader (BMG LabTech, Ortenberg, Germany) for 30 min. All values are given as fold of control of the 30 min value. H_2_O_2_ stimulation was used as control (positive control for DCFH-DA and DHR, negative control for DHE) [29]. 

#### 2.6.1. DCFH-DA

Dichloro-dihydro-fluorescein diacetate (H_2_DCFH-DA; Sigma, Darmstadt, Germany) was used at a concentration of 10 µM to determine overall ROS in the cells. Fluorescence was measured at Ex 485 nm/Em 520 nm. 

#### 2.6.2. Dihydrorhodamine 123 and Dihydroethidium

Dihydrorhodamine 123 (DHR) and dihydroethidium (DHE) were used to determine specific types of ROS: 10 μM DHR (Cayman Chemical, Ann Arbor, MI, USA) was used as an indicator for hydrogen peroxide, 10 µM DHE (Cayman Chemical) was used to measure superoxide anions. Fluorescence was measured at Ex 485 nm/Em 520 nm for DHR and Ex 544 nm/Em 590 nm for DHE.

### 2.7. Myeloperoxidase Activity

Myeloperoxidase (MPO) activity was measured with a myeloperoxidase activity kit II (Promocell, Heidelberg, Germany). Neutrophils were stimulated for 1 h at a concentration of 1 × 10^6^ cells/mL. Cells were lysed in assay buffer, and MPO activity was measured according to the manufacturer’s instructions. The protein content of the lysates was determined by the micro-Lowry technique, and the activity values were normalized to the protein content.

### 2.8. Neutrophil Elastase Activity

Neutrophil elastase (NE) activity was analyzed with the neutrophil elastase-specific substrate MeOSuc-AAPV-AMC (sc-201163, Santa Cruz Biotechnology, Heidelberg, Germany; [30,31]). Isolated neutrophils (1 × 10^6^ cells/mL) were stimulated with LPS (25 g/mL) or PMA (100 nM) ± insulin (160 IU/L) for 2 h. The supernatant of cells was collected and mixed with reaction solution (0.2 mM MeOSuc-AAPV-AMC in 0.1 M Tris pH 7.4) in a 1:1 ratio (final substrate concentration 0.1 mM). Change in fluorescence was measured over a time frame of 2 h at Ex 350–15 nm/Em 440–20 nm. The activity was determined as a slope from 30–70 min. 

### 2.9. Western Blot

Neutrophils were diluted to 1 × 10^6^ cells/mL and stimulated with PMA (100 nM) and/or insulin (160 IU/L) for 2 h. For the determination of cit-H3 levels, neutrophils were incubated for a total time of 3 h. After 2 h of incubation 0.5 mM PMSF were added to prevent cit-H3 degradation [32]. 

Cells were collected with a cell scraper and by subsequent centrifugation at 600× *g* for 10 min. The cell pellet was resuspended in RIPA buffer [33], kept on ice for 30 min, and then cell debris was removed by centrifugation at 13,000× *g* for 10 min at 4 °C. The protein content of the supernatant was determined by micro-Lowry. We loaded 35 µg of protein lysate onto a 12% SDS-PAGE gel and then transferred it to a nitrocellulose membrane through wet-blot transfer. Successful protein transfer was confirmed by Ponceau S staining and membranes were blocked with 5% BSA in TBS-T for 1 h. Antibody incubation was carried out overnight at 4 °C (Antibodies: MPO: Santa Cruz Biotechnology #52707; PAD4: Santa Cruz Biotechnology #365369; phospho-ERK: Cell Signaling Technology (Beverly, MA, USA) #4370; phospho-p38: Cell Signaling Technology #4511, cit-H3: Abcam #5103; HPRT: Santa Cruz Biotechnology #376938 was used as loading control). After washing with TBS-T, membranes were incubated with the respective horseradish peroxidase-coupled secondary antibodies (anti-mouse: Cell Signaling Technology #7076, anti-rabbit: Santa Cruz Biotechnology #2004) for 2 h and detection was carried out with enhanced chemiluminescence solution with an INTAS Chemocam (INTAS, Göttingen, Germany). 

### 2.10. Statistical Analysis

The groups were compared using the nonparametric Kruskal–Wallis test with Dunn’s correction for multiple comparisons or a Two-Way ANOVA mixed model for several variables. Statistical analysis was performed with Graphpad Prism 8 (San Diego, CA, USA). A *p*-value below 0.05 was considered significant. Data are shown as box plots with median, interquartile range, and 95% confidence interval (Tukey modification) where outliers are indicated as dots. For each experiment the number of replicates is given in the figure legend. “N” indicates the number of different donors per experiment, “n” indicates the number of technical replicates for each donor. 

## 3. Results

### 3.1. High Glucose and Insulin Did Not Induce NET Release

First, the basal effect of high glucose (HG) or high glucose with insulin (HG Ins) on spontaneous NET release was analyzed (Figure 2A–C). Neither HG nor HG Ins induced DNA release (Figure 2A) or NET formation (Figure 2B,C). 

To investigate possible priming of neutrophils by these diabetic conditions, cells were pre-incubated with HG Ins for 1 h and then stimulated with either PMA or CI, which use different pathways to activate NET release [32]. Both stimulants induced robust neutrophil DNA release (Figure 2D). Priming the cells with HG Ins had no effect on CI-induced DNA release, but slightly decreased PMA-induced total DNA release. Analysis of the NET formation dynamics (Figure 2E) showed that HG Ins significantly delayed DNA release following PMA treatment. Immunofluorescence staining of NET formation (Figure 2F) showed large extracellular structures staining positive for DNA and MPO after PMA treatment which are no longer present after the addition of HG Ins. Here, only decondensation of the chromatin is observed, but no DNA is released. HG itself did not stimulate neutrophils to release NETs, even in combination with PMA or CI or after a prolonged period of stimulation (Appendix A).

### 3.2. Insulin Delayed NET Formation

The observed effects of insulin on PMA-induced NET release were more closely investigated in the following. An extension of the observation time of DNA release by Sytox Green assay (Figure 3A) showed a clear temporal shift in the DNA release curve with the addition of insulin and PMA compared to PMA alone. However, the maximum amount of DNA released stayed the same. Analysis of the activation dynamics (Figure 3B) revealed a clear insulin-dependent delay (approximately 2 h) in the time to reach the half-maximal amount of NETs released. 

To confirm the observed effects, the experiments were repeated with different concentrations of insulin (10–640 IU/L, Figure 3C,D). Within this concentration range, insulin alone did not induce NET formation. The addition of insulin dose-dependently decreased the total amount of DNA released (Figure 3C) and NET formation induced by PMA treatment (Figure 3D). NET formation was significantly reduced compared with PMA alone starting at 160 IU/L. At this concentration, no significant differences in DNA release were detected compared with control conditions, despite PMA stimulation (Figure 3C). Increasing insulin concentrations led to the decondensation of chromatin within the cells, but no NET-like structures were observed outside the cells (Figure 3E). This was observed at insulin concentrations as low as 40 IU/L. Insulin alone did not show signs of NET release, even at the highest concentration of 640 IU/L (Figure 3E). 

### 3.3. Insulin and PMA Activated Cells but Reduced Cit-H3 Formation

Important prerequisites for the formation of NETs after PMA stimulation are the activation of mitogen-activated protein kinases (MAPKs) and ROS formation [34,35]. To find out if these factors are influenced by insulin treatment, the formation of ROS, MPO activity, and activation of MAPKs were analyzed. 

PMA strongly induced general ROS (Figure 4A). Insulin alone also strongly induced general ROS, leading to a cumulative effect with Ins+PMA, which reached nearly the level of the positive control hydrogen peroxide. Further specification of the type of ROS revealed a slight but not significant induction of superoxide radical and a strong induction of hydrogen peroxide by PMA alone. Insulin alone did not induce superoxide radical or hydrogen peroxide but seemed to slightly attenuate the effect of PMA (Figure 4B,C). Even after a longer stimulation time, cells without insulin stimulation did not reach ROS levels of insulin-stimulated cells (Appendix A). 

MPO activity was increased as expected by PMA and Ins+PMA but not by insulin incubation alone (Figure 4D). Ins+PMA incubation showed slightly lower MPO activities, however, the difference was non-significant. MAPK activation analyses showed strong phosphorylation of p38 and ERK after 2 h of stimulation with Ins+PMA whereas PMA alone did not activate these MAPKs at this time point. After 1 h of stimulation, only ERK was phosphorylated by PMA (Appendix A). After 3 h of stimulation and treatment with PMSF to prevent cit-H3 degradation, cit-H3 was strongly induced by PMA alone (Figure 4F, right). The presence of insulin prevented this increase. The levels of MPO or PAD4 were only slightly reduced with Ins+PMA compared to all other conditions which all had similar protein levels (Figure 4F, left and middle).

### 3.4. Insulin Also Delayed NET Formation by LPS 

PMA is known as a very strong but also an artificial inducer of NET formation in neutrophils [34]. To analyze the effect of insulin in more biologically relevant conditions, neutrophils were stimulated with LPS with or without insulin, and NET formation was analyzed. LPS stimulation induced robust neutrophil DNA release (Figure 5A). The addition of insulin shifted the start of DNA release to a later time point (Figure 5B). 

Comparable to the stimulation with PMA, stimulation with LPS and insulin-induced robust ROS induction (Figure 5C) whereas LPS alone did not induce ROS.

### 3.5. Insulin Modulates Pathogen Defense of Neutrophils

The results show that insulin strongly modulates NET formation. However, no information about the ability to fight pathogens is given. Neutrophil elastase (NE) released into the supernatant was quantified as a factor of the pathogen defense. NE activity in the supernatant of neutrophils was significantly increased by Ins+PMA. PMA alone showed only a slight but not significant increase in NE activity (Figure 6A). LPS and Ins+LPS induced similar levels of NE release whereas insulin alone did not induce NE release (Figure 6A).

Further, neutrophils were co-incubated with four different clinically relevant bacteria (SA: *S. aureus*; SE: *S. epidermis*; PA: *P. aeruginosa*; EF: *E. faecalis*) and NET formation and bacterial survival were analyzed. NET formation was only significantly induced by *S. aureus* and PMA and for both insulin addition significantly reduced the amount of formed NETs (Figure 6B). However, the microscopic images from Sytox Green assay show robust NET formation after *P. aeruginosa* stimulation at the 5 h timepoint (Figure 6C). *S. epidermis* and *E. faecalis* both did not induce relevant NET formation (Figure 6C). Enlarged images of the Sytox Green assay (Figure 6C lower row) show large NET structures after PMA stimulation whereas *P. aeruginosa* co-incubation led to smaller but many areas of NET formation, both with and without insulin. 

The analysis of bacterial survival after co-incubation with neutrophils revealed a significant reduction of *P. aeruginosa* survival after the addition of insulin (Figure 6D). For both *S. aureus* and *S. epidermis* the number of CFUs were reduced in tendency, but these differences were non-significant. 

## 4. Discussion

Neutrophils are strong mediators of inflammation and are the first line of defense against pathogens. Although neutrophils function as the first line of defense, a special mechanism of neutrophils, the release of extracellular neutrophil traps, has emerged as a problem in many diseases [36] Diabetics are particularly affected. Many efforts have been made to fine-tune the regulation of NET formation in pathological conditions. In this study, the effect of insulin on NET formation was analyzed. In contrast to previous studies [37,38], high glucose alone did not induce NET formation whereas insulin efficiently delayed NET formation induced by PMA or LPS but had no effect on NET formation induced by CI. Similarly, inhibition of NET formation in vitro was already shown for metformin [14] and liraglutide [17].

Mechanistically, insulin strongly induced ROS and MAPKs, both factors that are essential for NET formation [35,39] but did not induce NET formation itself. This suggests a regulatory role for insulin in NET formation. In an earlier study, insulin was found to have an immunosuppressive effect [40] by reducing superoxide radicals and nuclear factor kappa B (NFκB). In our study, a reduction in general ROS by insulin could not be observed; in contrast, ROS production was increased. Only a very slight increase in superoxide radical was observed after PMA ± insulin stimulation, this seems not to play a role here, in contrast to the above-mentioned study. Nitric oxide-related ROS forms such as the peroxynitrite anion are unlikely to be produced as they were shown before to induce NET formation in neutrophils [41], which we did not observe. Consequently, a shift in the type of ROS may contribute to the observed effects in our study.

In vivo, insulin was shown to induce bactericidal activity by induction of ROS and MPO expression in neutrophils from healthy volunteers [42]. However, it is likely that ROS plays a role in the insulin-mediated delay in NET formation as only LPS- or PMA-induced NET formation was influenced, but not the ROS-independent CI-mediated NET formation [32]. The overshooting ROS levels may inhibit PAD4 activity [43] which was shown here indirectly by reduced levels of cit-H3 after stimulation with PMA. PAD4 activity and the resulting cit-H3 formation are essential for NET formation, more clearly for the important chromatin decondensation [44]. Microscopic images clearly revealed differences in chromatin expansion by insulin, supporting the above-mentioned hypothesis that elevated ROS inhibit PAD4 activity. Furthermore, insulin slightly reduced MPO activity compared to PMA alone which could further decrease NET formation [45]. Figure 7 summarizes our findings for the different analyzed mediators of NET formation. 

Interestingly, a study in diabetic mice showed restoration of the bactericidal activity of neutrophils by insulin and therefore improvement of surgical site infections, which was explained by an increased phagocytotic activity induced by insulin and superoxide production [46]. We observed a similar effect after co-incubation of neutrophils with different bacteria: additional insulin reduced NET formation but without hampering the pathogen defense. In contrast, it even increased the killing of *P. aeruginosa*, an important opportunistic pathogen in the clinic [27]. Stimulation with *S. aureus* strongly induced NET formation, which is most likely induced by one of the many virulence factors of *S. aureus* such as phenol-soluble modulins (PSMs) [47]. NET formation induced by *S. aureus* has been shown to worsen skin infections [48], thus a reduction of NET formation by insulin may be beneficial for healing.

In addition, insulin application led to a systemic shift from an initial pro-inflammatory to an anti-inflammatory reaction in endotoxemic rats [49]. The observed effect was not related to a glucose-lowering effect of insulin [49]. A similar effect was observed in wound healing in diabetic rats where topical insulin application induced neutrophil phagocytosis and apoptosis and resulted in M2-polarization of macrophages. This immunomodulation improved the healing outcome [50]. Our data suggest that insulin triggers other defense mechanisms of neutrophils, which is supported by the above-mentioned studies. This would allow for tuning the balance between NET formation and other defense mechanisms in neutrophils, especially in cases where increased NET formation is known to be a problem. However, the effect of insulin on other cell types involved in (diabetic) wound healing was not investigated in this study. Earlier studies already demonstrated a positive effect of insulin on the migration abilities of keratinocytes [51] and less scar formation in the healing of rat burn wounds [52]. The impact of the insulin-mediated reduction of NET formation in a more complex setting for wound healing needs to be evaluated in further studies. Such studies could also include neutrophils from diabetic patients. In the present study neutrophils from healthy volunteers were used, which has to be seen as a second limitation of our study.

Diabetic neutrophils show generally dysregulated functions [53] and an altered NET formation in particular. They responded differently to various NET stimuli [54] and circulating NET markers are elevated in diabetics [15]. Since the effectiveness of pathogen defense by NETs is also reduced in T2DM patients [55], restoring bactericidal activity by insulin could be an interesting clinical target, which, however, still needs to be proven in a clinical setting. Our findings may explain previously seen positive effects on wound healing after topical application of insulin in diabetic rats [21] and support the observed altered polarization of macrophages [50]. Further, it may explain the finding of an improved survival of critically ill patients by strictly controlled insulin therapy [56].

## 5. Conclusions

In conclusion, insulin seems to suppress and delay the formation of NETs by different stimuli e.g., LPS and PMA but not calcium ionophore. Mechanistically, insulin increases ROS production, leads to phosphorylation of MAPKs and reduces cit-H3 formation after PMA stimulation. This general activation of the neutrophils results in increased clearance of *P. aeruginosa.* Systemic or local application of insulin could thus provide a new option to control overshooting NET formation in diabetic wound healing with possible additional effects such as restoration of bactericidal activity and, finally, improved healing.

## Figures and Tables

**Figure 1 biology-12-01082-f001:**
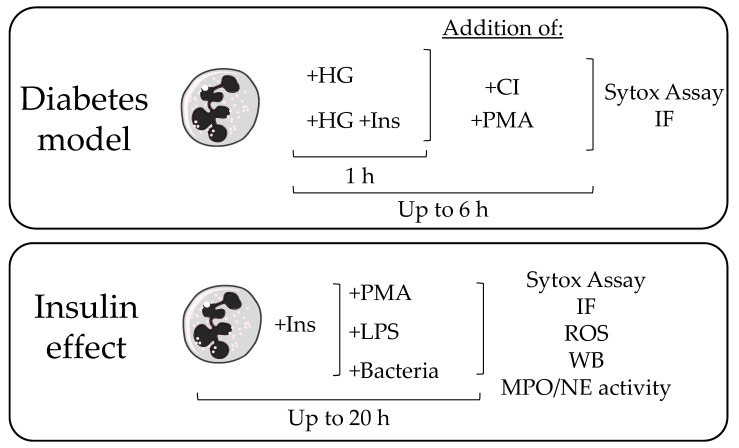
Overview of experimental set-up for the analysis of NET release in the diabetic model and to analyze the insulin effect. HG: high glucose (25 mM), Ins: insulin (160 IU/L), CI: calcium ionophore A23187, PMA: phorbol myristate acetate, IF: immunofluorescence, LPS: lipopolysaccharide, ROS: reactive oxygen species, WB: western blot, MPO: myeloperoxidase, NE: neutrophil elastase.

**Figure 2 biology-12-01082-f002:**
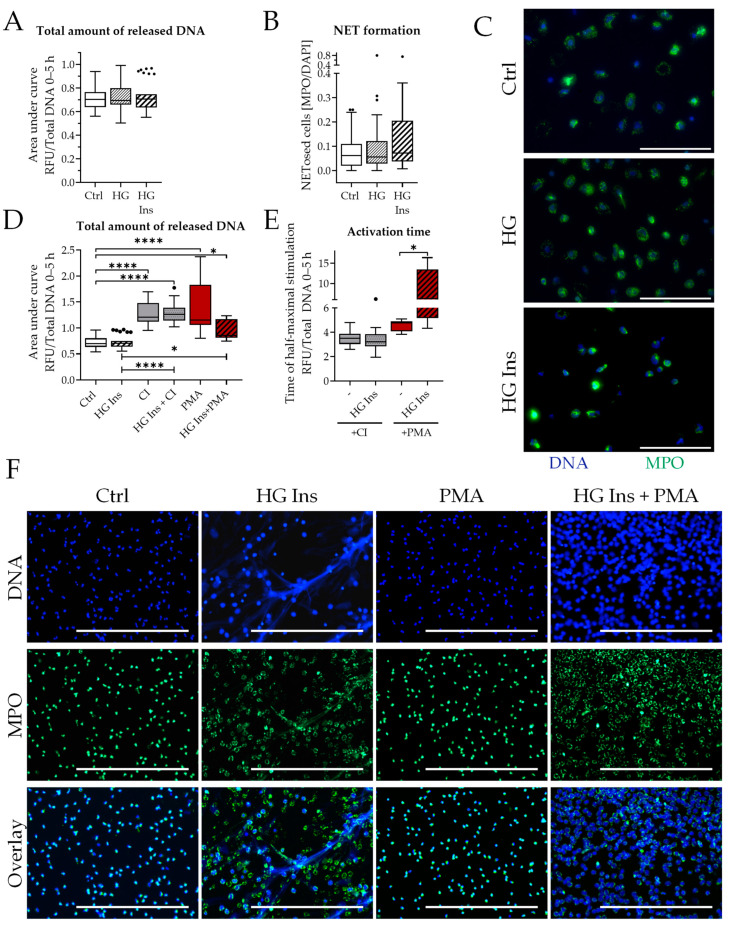
Analysis of NET formation in diabetic conditions (**A**) Total amount of released DNA determined by area under the curve from Sytox Green assay N = 5, n = 6. (**B**) NET formation was determined by immunofluorescence analysis after 3 h stimulation time. N = 7, n = 3. (**C**) Exemplary images of immunofluorescence. Scale bar 100 µm. Blue: DNA (Hoechst 33342), green: myeloperoxidase (MPO) (**D**) Total amount of released DNA determined by area under the curve from Sytox Green assay, 1 h pre-stimulation ± HG Ins, then addition of PMA/CI, total assay time 5 h. N = 5, n = 3. (**E**) Half-maximum stimulation time determined from Sytox Green assay, 1 h of pre-stimulation ± HG Ins, then addition of PMA/CI, total assay time 5 h. N = 5, n = 3. (**F**) Immunofluorescence staining of neutrophils after stimulation with PMA and HG Ins. Blue: DNA (Hoechst 33342), green: myeloperoxidase. Scale bar 400 µm. PMA: phorbol myristate acetate (100 nM), CI: calcium ionophore A23187 (4 µM). HG Ins: High glucose (25 mM) + Insulin 160 IU/L. * *p* < 0.05, **** *p* < 0.0001 as determined by Kruskal–Wallis test. Black dots indicate outliers.

**Figure 3 biology-12-01082-f003:**
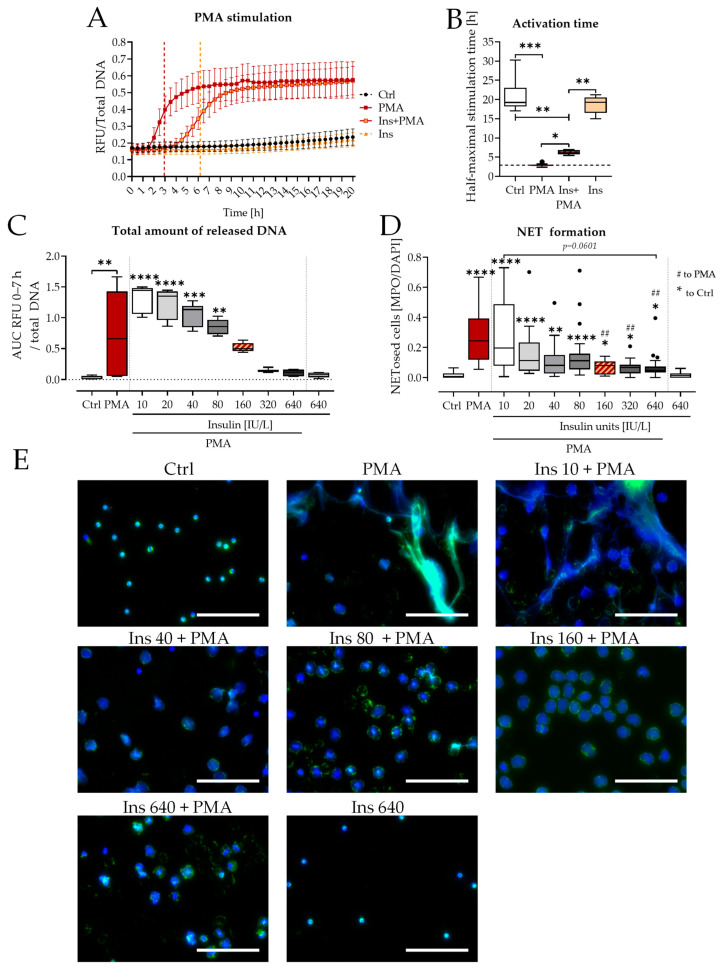
Insulin delays NET formation. (**A**) Time course of Sytox Green assay measurement 0–20 h. N = 4, n = 4. (**B**) Half-maximum stimulation time was determined from the Sytox Green assay. N = 4, n = 4. (**C**) Total amount of released DNA determined by Sytox Green assay after stimulation with different concentrations of insulin ± 100 nM PMA. N = 2, n = 3. (**D**) NET formation was determined by immunofluorescence analysis after stimulation with different concentrations of insulin ± 100 nM PMA (5 h). N = 4, n = 5. (**E**) Exemplary images of neutrophil immunofluorescence after stimulation with PMA and different concentrations of insulin (in IU/L). Blue: DNA (Hoechst 33342), green: myeloperoxidase. Scale bar 100 µm. PMA: 100 nM, Ins: Insulin 160 IU/L. * *p* < 0.05, ** *p* < 0.01, *** *p* < 0.001, **** *p* < 0.0001 as determined by Kruskal–Wallis test. ## indicates significance to PMA stimulated cells, *p* < 0.01. Black dots indicate outliers.

**Figure 4 biology-12-01082-f004:**
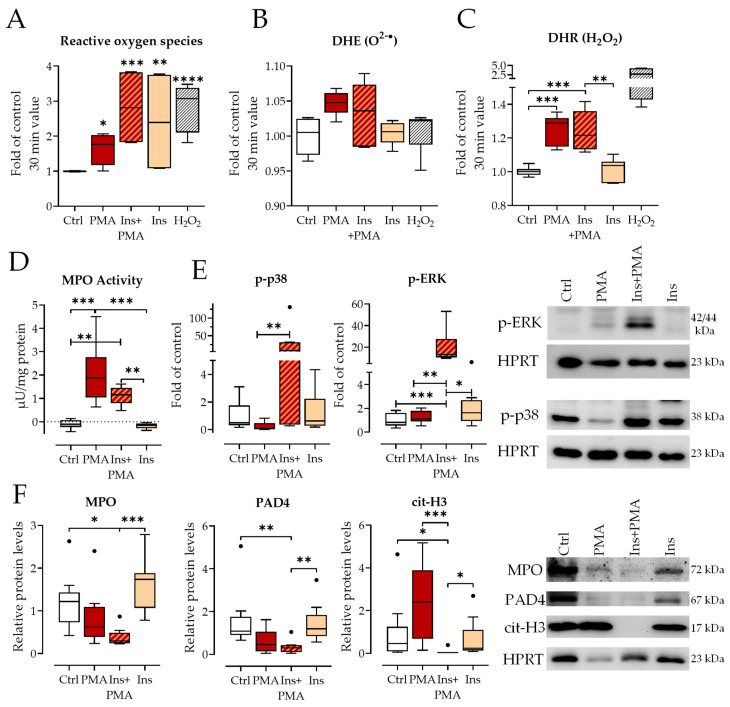
Insulin activates neutrophils. (**A**) Determination of total reactive oxygen species (ROS) by DCFH-DA assay. N = 4, n = 3. (**B**) Determination of superoxide anions by DHE (dihydroethidium) measurement. N = 3, n = 3. (**C**) Determination of hydrogen peroxide by DHR (dihydrorhodamine) measurement. N = 3, n = 3. (**D**) Analysis of MPO activity after 1 h stimulation. N = 4, n = 2. (**E**) Detection of phosphorylated MAPK levels (phospho-p38, left and phospho-ERK, right) by Western blot after 2 h stimulation of neutrophils. Exemplary images of bands are shown on the right. N = 4, n = 2. (**F**) Detection of MPO (left), PAD4 (middle), and citrullinated H3 (cit-H3, right) by Western Blot after 3 h of neutrophil stimulation (addition of 0.5 mM PMSF after 2 h). Exemplary images of bands are shown on the right. N = 6, n = 1–2. PMA: 100 nM, Ins: Insulin 160 IU/L, * *p* < 0.05, ** *p* < 0.01, *** *p* < 0.001,**** *p* < 0.0001 as determined by Kruskal–Wallis test. Black dots indicate outliers.

**Figure 5 biology-12-01082-f005:**
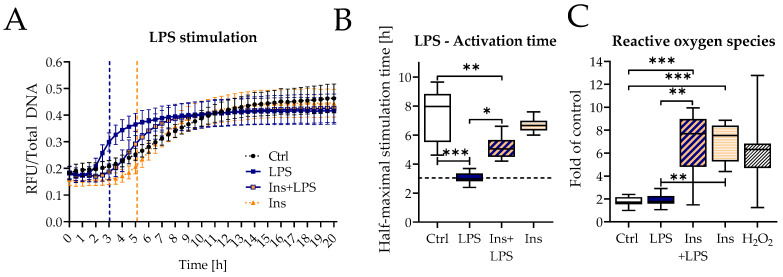
Effect of insulin on NET release by LPS. (**A**) Time course of Sytox Green assay measurement 0–20 h. N = 4, n = 4. (**B**) Half-maximum stimulation time was determined from the Sytox Green assay. N = 4, n = 4. (**C**) Determination of total ROS by DCFH-DA assay. N = 5, n = 3. LPS: lipopolysaccharide (25 µg/mL). Ins: Insulin 160 IU/L. * *p* < 0.05, ** *p* < 0.01, *** *p* < 0.001 as determined by Kruskal–Wallis test.

**Figure 6 biology-12-01082-f006:**
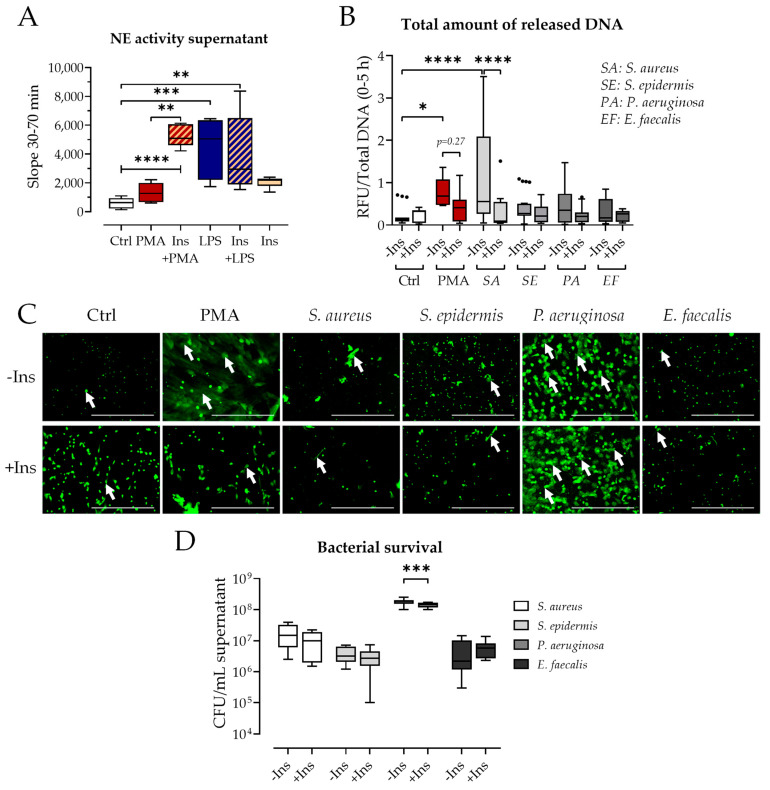
Effect of insulin on pathogen defense of neutrophils. (**A**) Neutrophil elastase (NE) activity N = 4, n = 2. (**B**) Total amount of released DNA determined by Sytox Green assay co-incubation with different bacteria (MOI 50) ± Ins N = 5, n = 3–4. (**C**) Exemplary images of Sytox Green staining after 5 h of stimulation. Green spots show extracellular DNA, white arrows indicate NET formation. The lowest row shows enlarged images of selected conditions. Scale bar 500 µm. (**D**) CFU after co-incubation of bacteria and neutrophils for 5 h. N = 5, n = 2. PMA: 100 nM, LPS: 25 µg/mL, Ins: Insulin 160 IU/L. * *p* < 0.05, ** *p* < 0.01, *** *p* < 0.001, **** *p* < 0.0001 as determined by Kruskal–Wallis test (**A**) or mixed-effect model (**B**,**D**). Black dots indicate outliers.

**Figure 7 biology-12-01082-f007:**
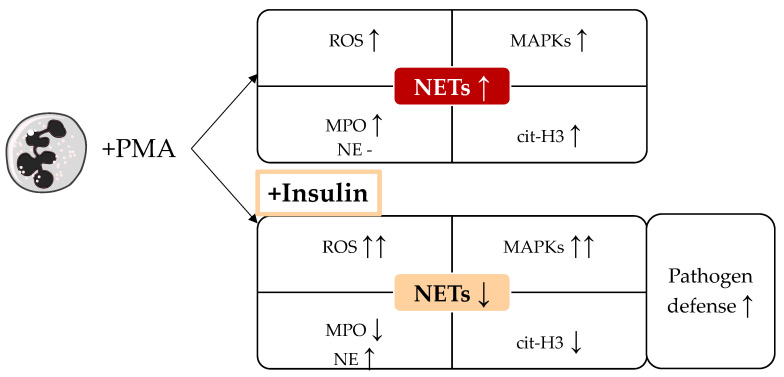
Summary of insulin effect on (PMA-induced) NET formation. ↑ indicates upregulation, ↓ indicates downregulation. PMA: phorbol myristate acetate, ROS: reactive oxygen species, MPO: myeloperoxidase, NE: neutrophil elastase, cit-H3: citrullinated histone H3, MAPKs: mitogen-activated protein kinases.

## Data Availability

The datasets generated during and/or analyzed during the current study are available from the corresponding author on reasonable request.

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
