# Peer review of "Insulin Can Delay Neutrophil Extracellular Trap Formation In Vitro—Implication for Diabetic Wound Care?"

_biology, 2023, doi:10.3390/biology12081082_

Round 1

Reviewer 1 Report

Article Number: biology-2505194

Article Title: Insulin can delay neutrophil extracellular trap formation in vitro - Implication for diabetic wound care?

In the article "Insulin can delay neutrophil extracellular trap formation in vitro - Implication for diabetic wound care?" the authors showed that in diabetic wound healing, Neutrophil extracellular traps (NETs), have been found to contribute to complications and are associated with amputations in patients with diabetic foot ulcers. This study showed that insulin effectively delayed the formation of NETs induced by certain stimuli (LPS and PMA). Mechanistically, insulin triggered reactive oxygen species, activated p38, and ERK signaling pathways, and reduced citrullination of histone H3. These findings suggest that insulin could be a valuable tool in regulating NET formation during diabetic wound healing. Because of the ever-increasing cases of diabetes worldwide, the study is of great importance.

Major Comments:

1.      It would be highly appreciated if the authors could provide high-resolution and magnified images for all panels in the manuscript. Additionally, including phase pictures to demonstrate cell morphology would be beneficial.

2.      For Figure 6C, it is kindly requested to include colored, magnified, and high-resolution pictures to enhance clarity and visibility.

3.      Supplementary materials should be treated as an integral part of the manuscript. To ensure consistency, please ensure that figures in the supplementary materials are of comparable quality to those in the main article. It would be helpful to mark the lanes and provide figure legends clearly. Moreover, providing additional information (such as numbers) about the supplementary materials within the main manuscript would be highly valuable.

4.      Please remember to include molecular weight markers in all gel images, including those in the supplementary materials. This will aid in accurate interpretation and analysis.

5.      The conclusion section appears to be too concise to comprehend the findings fully. We kindly request the authors to revise and expand the conclusion, providing a more comprehensive summary of the results. Additionally, further discussion of the implications and significance of the findings would be greatly appreciated.

6.      It is recommended that the authors include more recent studies and remove outdated references to ensure the manuscript reflects the most current and relevant research in the field.

Minor Comments:

1.      It would greatly enhance the manuscript's clarity if terminologies such as PMA, LPS, MPO, etc., are defined when first introduced. This will ensure that readers can easily understand the concepts being discussed.

2.      Please be advised that the manuscript has several grammatical and typographical errors. It would be beneficial to carefully review and correct these errors to maintain the overall quality of the writing.

3.      It is kindly requested to mention the corresponding figure numbers at their respective places in the text. For example, mentioning "Figure 1" at the appropriate location in the text will assist readers in locating the relevant figure.

4.      To maintain consistency, it is recommended to adhere to a uniform reference formatting style throughout the manuscript. Ensuring that the references follow the same style will contribute to the overall professional appearance of the article.

5.      The conclusion section appears to be insufficient in providing a comprehensive understanding of the main findings. It would be appreciated if the conclusion could be rephrased and expanded upon, elaborating on the key outcomes of the study and their significance in a more detailed manner.

Language improvement is required. 

Reviewer 2 Report

Dear Author's

Thank you for the opportunity to read the text of the manuscript.
I suggest some minor additions, namely: please complete Limitations of the study
and some cited references is downright historic...!

best regards

Author Response

Dear Author's
Thank you for the opportunity to read the text of the manuscript. I suggest some minor additions, namely: please complete Limitations of the study and some cited references is downright historic...! best regards

Re: We would like to thank the reviewer for the valuable comment. We replaced the outdated references and added some limitations of the study to the discussion.

Reviewer 3 Report

The purpose of this study was to look into how insulin affects NET development. The development of NET after activating neutrophils with various agents (such as PMA, LPS, or calcium ionophore) in the presence or absence of insulin was examined by the researchers. The findings demonstrated that whereas insulin had no effect on NET development produced by calcium ionophore, it dramatically slowed NET formation generated by LPS and PMA. These results imply that insulin, either systemically or topically administered, may be a possible strategy for controlling NET development during diabetic wound healing. I enjoyed reading this manuscript and think that this discovery is significant for diabetes research. The writing in the manuscript is compelling and persuasive. Here are my comments:

1.       Did the authors assess if particular insulin receptor subtypes may be involved in the effects of insulin on NET formation? Please include an experiment by modulate insulin signaling with insulin receptor agonists or antagonists and assess the effects on NET formation.

2.       Assessing the impact of insulin on extracellular matrix deposition, fibroblast proliferation, angiogenesis and other crucial elements of wound healing can strengthen the correlation between insulin and NET formations. Did the authors examine whether changes in these wound healing parameters are correlated with insulin's control of NET formation?

3.       Please include an experiment to examine the potential effects of insulin versus other widely prescribed anti-diabetic medications, such as Metformin and Liraglutide, in controlling NET development.

4.       Please provide magnified images for all the immunofluorescence images. Magnified images of the cells/staining may help in better interpreting the results.  
